# Metabolic Signatures in Coronary Artery Disease: Results from the BioHEART-CT Study

**DOI:** 10.3390/cells10050980

**Published:** 2021-04-22

**Authors:** Stephen T. Vernon, Owen Tang, Taiyun Kim, Adam S. Chan, Katharine A. Kott, John Park, Thomas Hansen, Yen C. Koay, Stuart M. Grieve, John F. O’Sullivan, Jean Y. Yang, Gemma A. Figtree

**Affiliations:** 1Cardiothoracic and Vascular Health, Kolling Institute, Northern Sydney Local Health District, Sydney, NSW 2065, Australia; sver4267@sydney.edu.au (S.T.V.); owen.tang@sydney.edu.au (O.T.); katharine.kott@sydney.edu.au (K.A.K.); johnpark.1997@outlook.com (J.P.); Tommi_hansen1@hotmail.com (T.H.); 2Department of Cardiology, Royal North Shore Hospital, Sydney, NSW 2065, Australia; 3Northern Clinical School, Faculty of Medicine and Health, University of Sydney, Sydney, NSW 2006, Australia; 4Charles Perkins Centre, University of Sydney, Sydney, NSW 2006, Australia; taiyun.kim@sydney.edu.au (T.K.); adam.s.chan@sydney.edu.au (A.S.C.); Yen.Koay@hri.org.au (Y.C.K.); john.osullivan@sydney.edu.au (J.F.O.); jean.yang@sydney.edu.au (J.Y.Y.); 5School of Mathematics and Statistics, University of Sydney, Sydney, NSW 2006, Australia; 6Computational Systems Biology Group, Children’s Medical Research Institute, Westmead, NSW 2145, Australia; 7Central Clinical School, Faculty of Medicine and Health, University of Sydney, Sydney, NSW 2006, Australia; 8Heart Research Institute, The University of Sydney, Sydney, NSW 2042, Australia; 9Imaging and Phenotyping Laboratory, Charles Perkins Centre, Faculty of Medicine and Health, University of Sydney, Sydney, NSW 2006, Australia; stuart.grieve@sydney.edu.au; 10Department of Radiology, Royal Prince Alfred Hospital, Sydney, NSW 2050, Australia

**Keywords:** coronary artery disease, metabolites, risk factors, atherosclerosis, metabolomics

## Abstract

Despite effective prevention programs targeting cardiovascular risk factors, coronary artery disease (CAD) remains the leading cause of death. Novel biomarkers are needed for improved risk stratification and primary prevention. To assess for independent associations between plasma metabolites and specific CAD plaque phenotypes we performed liquid chromatography mass-spectrometry on plasma from 1002 patients in the BioHEART-CT study. Four metabolites were examined as candidate biomarkers. Dimethylguanidino valerate (DMGV) was associated with presence and amount of CAD (OR) 1.41 (95% Confidence Interval [CI] 1.12–1.79, *p* = 0.004), calcified plaque, and obstructive CAD (*p* < 0.05 for both). The association with amount of plaque remained after adjustment for traditional risk factors, ß-coefficient 0.17 (95% CI 0.02–0.32, *p* = 0.026). Glutamate was associated with the presence of non-calcified plaque, OR 1.48 (95% CI 1.09–2.01, *p* = 0.011). Phenylalanine was associated with amount of CAD, ß-coefficient 0.33 (95% CI 0.04–0.62, *p* = 0.025), amount of calcified plaque, (ß-coefficient 0.88, 95% CI 0.23–1.53, *p* = 0.008), and obstructive CAD, OR 1.84 (95% CI 1.01–3.31, *p* = 0.046). Trimethylamine N-oxide was negatively associated non-calcified plaque OR 0.72 (95% CI 0.53–0.97, *p* = 0.029) and the association remained when adjusted for traditional risk factors. In targeted metabolomic analyses including 53 known metabolites and controlling for a 5% false discovery rate, DMGV was strongly associated with the presence of calcified plaque, OR 1.59 (95% CI 1.26–2.01, *p* = 0.006), obstructive CAD, OR 2.33 (95% CI 1.59–3.43, *p* = 0.0009), and amount of CAD, ß-coefficient 0.3 (95% CI 0.14–0.45, *p* = 0.014). In multivariate analyses the lipid and nucleotide metabolic pathways were both associated with the presence of CAD, after adjustment for traditional risk factors. We report novel associations between CAD plaque phenotypes and four metabolites previously associated with CAD. We also identified two metabolic pathways strongly associated with CAD, independent of traditional risk factors. These pathways warrant further investigation at both a biomarker and mechanistic level.

## 1. Introduction

Large longitudinal epidemiological studies have identified a number of modifiable risk factors and disease processes associated with cardiovascular disease (CVD), including hypertension, hyperlipidaemia, cigarette smoking, and diabetes mellitus [1]. Public health and primary prevention programs targeting these modifiable risk factors have substantially reduced the burden of disease at the community level [2]. Despite this, coronary artery disease (CAD) remains the leading cause of death globally and at an individual level it is not uncommon for people to suffer life threating cardiovascular events such as ST elevation myocardial infarction in the absence of these well described risk factors [3,4,5,6,7]. Clinically utilised biomarkers, including standard lipid profiles, blood glucose levels, and HbA1c, reflect established traditional risk factors or CAD, however we lack biomarkers that reflect the residual risk not captured by traditional risk factors.

Computed tomography coronary angiography (CTCA) can non-invasively assess and characterise coronary artery plaque, as well as clearly identify normal coronary arteries (the absence of plaque). This provides a powerful research opportunity for CAD biomarker discovery studies, overcoming a challenge faced by many prior studies that relied on cardiovascular events to define CAD, where many of the healthy control participants actually have substantial subclinical atherosclerosis [8,9,10] BioHEART-CT, a multicentre, prospective cohort study of patients with suspected CAD, utilises detailed demographic, clinical, and CTCA data to accurately quantify the disease burden in its participants, and stored blood samples for molecular phenotyping. It has been designed to identify novel biomarkers associated with CAD in order to improved risk stratification and to identify new mechanistic pathways that could be targeted to reduce the burden of CAD [11].

Metabolites are influenced by environmental factors, genetic factors, and interactions between the two [12,13]. Metabolomics, the systematic identification and quantification of small molecules (up to 1.5 kDa) in biological fluids, thus reflects the metabolic state of the biological system at a point in time [14]. The metabolome can be assessed at a cellular, tissue, organ, or whole-organismal level. A recent systematic review of metabolomics studies attempting to identify metabolic signatures of incident cardiovascular disease identified 5 metabolites acylcarnitine, dicarboxylacylcarnitine, trimethylamine N-oxide (TMAO), phenylalanine, and glutamate [15]. Another recent study identified an independent association between dimethylguanidino valerate (DMGV) and incident CAD [16]. We hypothesised that there are circulating metabolites and shifts in particular metabolic pathways associated with the presence of CAD, and specific CAD phenotypes, including calcified plaque and non-calcified “soft” plaque. We utilised a large, well phenotyped cohort of stable patients with suspected CAD undergoing CTCA, and considered the role of traditional CAD risk factors. We proceeded to test these hypotheses in the BioHEART CT study, utilising liquid chromatography mass spectrometry.

## 2. Methods

### 2.1. BioHEART-CT Study

BioHEART-CT is a multicentre, longitudinal, prospective cohort study of patients undergoing a CTCA for investigation of suspected CAD. BioHEART is a registered Australian and New Zealand Clinical Trial (ACTRN12618001322224). The study protocol and design has been approved by the Northern Sydney Local Health District Human Research Ethics Committee (HREC/17/HAWKE/343) and described in detail previously [11]. Informed written consent was obtained. Patient level information was collected using a structured data collection form at the time of recruitment and entered into a secure, encrypted database. Baseline data collected includes demographics, anthropometric measures, past medical diagnoses (cardiovascular diagnoses and risk factors), current medication use, and social history. Standard Modifiable cardiovascular Risk Factors (SMuRFs) were recorded based on the self-reported history or use of pharmacotherapy for any of hypertension, hypercholesterolaemia, diabetes mellitus, or a significant smoking history. Current smoking was defined as having regularly smoked within the past 12 months, and significant smoking history was considered to be 10 packet years or greater. A significant family history of premature ischaemic heart disease in a first degree relative under the age of 60 years was recorded. Blood samples were taken at the time of recruitment and stored in a biobank. Biochemical parameters including fasting cholesterol profiles, fasting blood glucose, or HbA1c were not available for this analysis. CTCA imaging data was obtained and securely stored to allow detailed phenotypic analysis [11]. The inclusion criteria were age 18 years or older referred for clinically indicated CTCA for investigation of suspected CAD, willing and able to provide informed consent. The exclusion criteria were patients who are highly dependent on medical care who were unable to provide informed consent, patients who were unwilling or unable to participate in ongoing telephone-based follow-up. The BioHEART-CT discovery cohort assessed in this study includes the first 1002 patients recruited to the BioHEART-CT registry who had a technically adequate CTCA, sufficient stored blood samples for all the planned biomarker discovery platforms and who did not have a cardiac stent in situ or a prior history of coronary artery bypass surgery.

### 2.2. CTCA

CTCA scans were performed on a 256 slice Philips CT scanner using standard clinical protocols, as per supervising radiologists with at least level two RANZCR accreditation. If required, oral metoprolol or ivabradine was utilised to achieve acceptable heart rate control. Sub-lingual nitro-glycerine was administered immediately prior to image acquisition to ensure adequate vasodilation. If the heart rate was sufficiently controlled image acquisition was performed prospectively, otherwise retrospective acquisition was utilised. Radiation doses were minimised in line with current recommendations [17,18]. Coronary artery reconstructions were performed using vendor software.

### 2.3. CTCA Disease Severity Scores

CAD burden was assessed using: a coronary artery calcium score (CACS), a 17 segment Gensini stenosis score and a 17 segment soft plaque score [19,20].

### 2.4. Coronary Artery Calcium Score

In brief, hyperattunated areas of at least 1 mm^2^ or ≥ 3 adjacent pixels with >130 Hounsfield units (HU) were incorporated into the CACS using the well described and validated Agatston method [20].
(1)$$\underset{1\;\leq \;x\;\leq 17}{\sum }\;Attenuaton\;area*denisty\;significance\;multiplier\left(x\right)$$

### 2.5. Gensini Score

The Gensini score is a validated CAD severity score [19]. Each of the 17 segments, as defined by the Society of Cardiovascular Computed Tomography 17-segment model, were assessed for degree of stenosis (*x*) (0%, 0–24%, 25–49%, 50–69%, 70–99%, 100%), and multiplied by a segmental significant factor representing both the proximity of the segment and the relative importance of the vessel segment (*y*) as listed in Figure 1 [21].
(2)$$\underset{1\;\leq \;x\;\leq 17}{\sum }\;stenosis\;score\left(x\right)*segmental\;significance\;multiplier\left(y\right)$$

### 2.6. Soft Plaque Score (SPS)

The plaque composition present in each segment with disease was also assessed and a plaque composition multiplier (*z*) (soft only ×3, mixed ×2, calcified only ×1) was recorded. Semi-quantitative assessments of plaque morphology are reproducible; additionally, soft plaque identified on CTCA may be associated with high-risk plaque morphology such as thin-cap fibroatheroma, disease progression, and adverse prognosis [22,23,24,25]. By incorporating a plaque composition multiplier for each scored segment we were able to assess the soft (biologically active) plaque burden. This results in a score reflecting the non-calcified or “soft” plaque burden, we defined as the Soft Plaque Score (SPS).
(3)$$\left[\underset{1\;\leq \;x\;\leq 17}{\sum }\;\left(stenosis\;score\left(x\right)*segmental\;significance\;multiplier\left(y\right)*plaque\;composition\;mutiplier\left(z\right)\right)\right]-Gensini\;score$$

Additionally, individuals with ≥1 stenosis with >50% stenosis were considered to have obstructive CAD if they had one or more >50% stenosis.

### 2.7. Plasma Sample Processing and Extraction Procedure

20–30 mls of venous blood was collected at the time of CTCA. Patients were advised to fast for 2 h prior to the procedure. The blood was processed, aliquoted, and stored at −80 °C. For metabolomic assessment, thawed plasma samples were prepared and analysed with slight modifications as previously described [26,27]. In brief, plasma samples were deproteinized using acetonitrile/methanol/formic acid (75:25:02; *v/v/v*) containing deuterated internal standards 10 mM valine-d_8_ (98%; Sigma-Aldrich, St Louis, MS, USA) and 25 mM phenylalanine-d_8_ (98%; Cambridge Isotope Laboratories, Inc., Tewksbury, MA, USA) for Hydrophilic Interaction Chromatography (HILIC) on an Atlantis^®^ HILIC column, and acetonitrile/methanol (75:25; *v/v*) containing 10 mM thymine-d_4_, 10 mM L-phenylalanine-d_8_ (98%; Cambridge Isotope Laboratories, Inc. Tewksbury, MA, USA). After vortexing, the samples were centrifuged at 20,000× *g* at 4 °C for 15 min, and the supernatant was transferred to HPLC-grade glass vials with inserts (Waters).

### 2.8. Liquid Chromatography Mass Spectrometry

Targeted metabolomic analysis measured hydrophilic metabolites in positive and negative ionization mode using an LC-MS system comprised of an Agilent 1260 Infinity liquid chromatography (LC) system coupled to a QTRAP 5500 mass spectrometer (MS) (AB SCIEX). The resulting metabolite containing supernatant was resolved on Agilent 1260 Infinity HPLC System and m/z determined by Qtrap5500 (Sciex) [28,29]. A hydrophilic interaction liquid chromatography (HILIC) tandem mass spectrometry (LC-MS/MS) method was used for the simultaneous detection of polar metabolites in positive ionization mode, composed of amino acids, nucleotides, neurotransmitters, and vitamins [30,31].

### 2.9. Data Processing and Normalisation

The raw data files (Analyst software, version 1.6.2; AB Sciex, Foster City, CA, USA) were imported into the built for purpose vendor analysis software Multi-Quant 3.0 for MRM Q1/Q3 for peak integration. Mass divided by charge (m/z) signals were obtained using the area under the curve for calibrated peaks from MultiQuant (SCIEX) with manual adjustments to the curves where necessary. 100 metabolites were measured, and 53 metabolites were detected at adequate levels in plasma to be included in the analyses.

We used a novel hierarchical normalisation approach for the removal of unwanted variation (hRUV) that utilises individual sample level replicates within and between batches for estimation of unwanted variation in the data, rather than depending solely on pooled QC sample replicates^32^. This method is superior to traditional normalisation techniques at removing technical variation whilst retaining biological signal [32].

### 2.10. Candidate Metabolites

4 metabolites (DMGV, glutamate, phenylalanine, and TMAO) in our panel of 53 were initially examined as candidate biomarkers based on prior studies, suggesting association with CAD and/or a biological role in atherosclerosis.

### 2.11. Total Metabolite Analysis

We assessed for associations metabolites between each of the 53 metabolites and the CAD scores. Given the unbiased nature of the analyses, *p*-values were adjusted, controlling for a false discovery rate (FDR) of five percent [33].

### 2.12. Metabolic Pathways

Each of the metabolites were categorised according to the primary metabolic pathways (super-pathways and sub-pathways) they are involved in, based on prior literature (Appendix A). Super pathways with ≥4 and ≤10 metabolites were assessed for association with the presence of CAD. As the amino acid super pathway had 32 metabolites, sub-pathways of this super-pathway that had ≥4 metabolites were assessed.

### 2.13. Statistical Analysis

Categorical variables are summarized using frequencies and percentages and numeric variables are summarized using mean, standard deviation, median, and interquartile ranges. Binary logistic regression models to assess associations with positive CAD-scores were performed against control groups (Table 1). The CAD severity scores, including CACS, Gensini and SPS, were right skewed due to a large proportion of zero scores. As previously described, they are not readily amenable to normalising transformations [34,35,36,37]. After the removal of zero scores, the log2 transformed CAD severity scores were approximately normal. We performed linear regression models on the log2 transformed severity scores to assess association with plaque burden. The covariates included in multivariable models were sex, body mass index (BMI), hypertension, significant smoking history, diabetes mellitus, and hypercholesterolaemia. Benjamini and Hochberg adjusted *p*-values, controlling for a FDR of five per cent, are presented for the unbiased analyses [33]. We used Multivariate analysis of variance (MANOVA) and Multivariate analysis of covariance (MANCOVA) to assess for association between the presence of CAD and metabolic pathways, using the MANOVA function and the default test statistic (Pillai–Bartlett statistic) in R. Statistical analyses were performed in R version 4.0.3 [38].

## 3. Results

Patient demographics, risk factors, and regular medication use are presented in Table 1. Overall, 66% of the cohort had CAD, 59% had calcified plaque, 54% had non-calcified (soft) plaque, and 18% had obstructive CAD present (Table 1). The median age of the cohort was 62 years, and those with CAD were older than those with no CAD (65 years [interquartile range 58–72] vs. 54 years [IQR 46–60]). Fourty-five percent of the cohort was female, with no substantial differences between those with or without CAD or other CAD parameters (Table 1). CAD patients were more likely to have standard cardiovascular risk factors, frequently multiple co-existent risk factors. Each of the standard cardiovascular risk factors were more common in those with CAD, and those with CAD were more likely to have multiple risk factors (Table 1).

### 3.1. Candidate Metabolite Associations with CAD

DMGV, glutamate, phenylalanine, and TMAO were initially examined as candidate biomarkers based on prior studies suggesting association and biological role in atherosclerosis. These analyses are presented in Table 2.

In univariate analysis, DMGV was associated with presence of CAD, Odds Ratio (OR) 1.41 (95% Confidence Interval [CI] 1.12–1.79, *p* = 0.004), presence of calcified plaque, OR 1.59 (95% CI 1.26–2.01, *p* = 0.0001), presence of soft plaque, OR 1.40, (95% CI 1.11–1.75), *p* = 0.004), and presence of obstructive CAD, OR 2.33 (95% CI 1.59–3.43, *p* < 0.0001). DMGV was also associated with the amount of CAD, ß coefficient 0.30 (95% CI 0.14–0.45, *p* = 0.0002) and amount of calcified plaque, ß coefficient 0.49 (95% CI 0.11–0.87, *p* = 0.011). The association with the amount of CAD remained when adjusted for age, sex, BMI, and traditional risk factors, ß coefficient 0.17 (95% CI 0.02–0.32, *p* = 0.026). We performed post-hoc mediation analyses to evaluate for interactions between traditional risk factors for CAD and DMGV. DMGV was strongly correlated with diabetes mellitus, hypertension, BMI, and age (*p* < 0.001 for all), and also correlated with smoking (*p* = 0.01) (Appendix A). In separate multivariable univariate analyses with DMGV and each of the traditional risk factors (one at a time), DMGV remained significantly associated with CAD when adjusted individually for each of the risk factors other than age and BMI (Appendix A).

Glutamate was associated with the presence of non-calcified plaque in univariate analysis, OR 1.48 (95% CI 1.09–2.01, *p* = 0.011), however glutamate was not associated with any of the other CAD measures assessed. In univariate analyses, phenylalanine was associated with amount of CAD, ß coefficient 0.33 (95% CI 0.04–0.62, *p* = 0.025), amount of calcified plaque, ß coefficient 0.88 (95% CI 0.23–1.53, *p* = 0.008), and presence of obstructive CAD, OR 1.84 (95% CI 1.01–3.31, *p* = 0.046). The association with amount of CAD and amount of calcified plaque remained when adjusted for age, sex, BMI, and traditional risk factors, ß coefficient 0.26 (95% CI 0.01–0.52, *p* = 0.046), and ß coefficient 0.77 (95% CI 0.19–1.35, *p* = 0.01), respectively.

TMAO was negatively associated with the presence of soft plaque in univariate and multivariable analyses (adjusting for age, sex, BMI, and traditional risk factors), OR 0.72 (95% CI 0.53–0.97, *p* = 0.029), and OR 0.65 (95% CI 0.46–0.92, *p* = 0.016), respectively. There was also a negative association between TMAO and with total amount of plaque and amount of calcified plaque in the multivariable models, ß coefficient −0.21 (95% CI −0.42–−0.01, *p* = 0.044) and ß coefficient −0.58 (95% CI −1.05–−0.10, *p* = 0.017). These four metabolites were also included in the unbiased metabolomic analyses however many of the associations did not retain significance when assessing all 53 metabolites and adjusting for an FDR of 0.05 (Appendix A).

### 3.2. Metabolomic Analyses

The pre-specified unbiased univariate and multivariable associations between the metabolites and disease phenotypes are presented in Figure 2, Appendix A. The top ranked metabolites associated with the presence of CAD were cAMP, uridine, DMGV, proline, asparagine, spremine, 2-arachidonyl glycerol, adenosine, thymidine, thiamine, and kynurenic acid (Figure 2). These associations were not statistically significant when controlling for a false discovery rate of five per cent.

However, even when controlling for a five percent FDR, DMGV remained strongly associated with the presence of calcified plaque, OR 2.33 (95% Confidence Interval [CI] 1.59–3.43, FDR adjusted *p* value = 0.0009) and obstructive CAD, OR 1.68 (95% CI 1.26–2.25, FDR adjusted *p* value = 0.025), Figure 1. DMGV was also associated with the amount of CAD present as assessed by Gensini score (ß coefficient 0.3, 95% CI 0.14–0.45, FDR adjusted *p* value = 0.014, Appendix A). Additionally, proline, spermine, uridine, cAMP, and kynurenic acid were all significantly associated with presence of calcified plaque (FDR adjusted *p* value < 0.05 for all, Figure 1). However, these associations did not remain after adjustment for traditional cardiovascular risk factors in the pre-specified multivariable models.

### 3.3. CAD Associations with Metabolic Pathways

Presence of CAD was strongly associated with the nucleotide metabolic pathway (comprised of: adenosine, cAMP, 2’-deoxyadenosine, cytosine, thymidine, and uridine) in MANOVA analysis, F statistics with degrees of freedom denoted as F(df1, df2), F(6, 995) = 3.903, *p* = 0.0007, Pillai = 0.023. This association remained when adjusted for age, gender and standard cardiovascular risk factors F(6, 980) = 3.933, *p* = 0.0007, Pillai = 0.024. CAD was also significantly associated with the lipid metabolic pathway (comprised of: carnitine, anadamide, arachidonic acid, butrylcarnitine, acetylcarnitine, 2-arachidonyl glycerol, physphocholine, choline, and TMAO), F(9, 992) = 2.008, *p* = 0.035, Pillai = 0.018. This association remained when adjusted for age, gender and standard cardiovascular risk factors F(9, 997) = 2.013, *p* = 0.035, Pillai = 0.018. There were no statistically significant associations between CAD and the leucine/isoleucine/valine metabolic pathway or the tryptophan metabolic pathway in multivariate analyses.

## 4. Discussion

Mass spectrometry platforms allow comprehensive assessment of the metabolites in a biological fluid at a point in time (metabolomics). Utilising a large cohort with CT coronary angiography characterisation of coronary atherosclerosis burden, we report novel associations between CAD plaque phenotypes and four candidate metabolites (TMAO, DMGV, glutamate, and phenylalanine) [15]. DMGV was most strongly associated with CAD and with more CAD plaque features.

In our study plasma TMAO was negatively associated with the presence of soft plaque, total plaque burden, and calcified plaque burden in univariate and multivariable models after adjusting for age, sex, and traditional risk factors. Whilst this finding contrasts with some prior studies that have found TMAO is associated with increased incidence of major adverse cardiovascular events in patients with established CAD, peripheral artery disease or chronic kidney disease, it is in keeping with other studies that have not confirmed an association with CAD and have suggested there may be a more nuanced relationship [39,40,41]. TMAO is generated through the oxidation of trimethylamine (TMA) by TMA monooxygenase present in some gut microbiota [42]. TMA is a downstream metabolite of dietary betaine, L-carnitine, and choline, which are particularly prevalent in red meats and eggs. In vitro and in vivo studies have demonstrated an association with enhanced atherosclerosis, thrombosis, and inflammation [41]. Mechanistic work is ongoing; however, current literature suggests choline, TMAO, and betaine have an atherogenic effect through the upregulation of macrophages and formation of foam cells, reduced cholesterol clearance, proinflammatory cytokines (including tumour necrosis factor-α and interleukin-1β), enhanced endothelial and smooth muscle cell leukocyte adhesion, and enhanced platelet reactivity [42,43]. Bordoni et al. recently reported that the association between TMAO/TMA levels and CAD is dependent on the rs247616 (cholesterol ester transfer protein) genotype. Individuals with the rs247616-CC genotype that had CAD had higher levels of TMAO compared to individuals with the same genotype but no CAD [39]. These findings highlight the intricacies of gene and environment interactions on the development of CAD and the need for personalised approaches to CAD risk assessments.

We also identified two metabolic pathways associated with CAD independently of traditional risk factors: the nucleotide metabolic pathway and the lipid metabolic pathway. The association of these metabolic pathways with atherosclerosis may reflect a mechanistic role in disease development or susceptibility, or potentially reflect the disease activity and host response to the disease itself. Important factors to consider are the known biology related to the metabolites and disease, as well as prior clinical data.

Plasma DMGV levels have previously been shown to be associated with incidence of type 2 diabetes mellitus (T2DM) and liver fat [27]. A subsequent study confirmed the association between DMGV and incident T2DM and identified an association with incident CAD [16]. Whilst DMGV levels were also associated with BMI and fasting triglycerides, an association with CAD remained after adjusting for traditional cardiovascular risk factors [16]. In our analyses, whilst DMGV was associated with calcified plaque, obstructive CAD, and amount of CAD, the association did not remain after adjustment for traditional risk factors. In post-hoc mediation analyses we identified strong univariate associations between diabetes, hypertension, BMI, and age with DMGV (*p* < 0.001 for all). DMGV remained associated with CAD when adjusted individually for each of the risk factors other than age and BMI, consistent with prior literature and suggesting that DMGV may be a circulating marker of the metabolic syndrome, with dysregulated DMGV homeostasis preceding cardiometabolic disease [16]. Alanine glyoxylate aminotransferase II (AGXT2) knock out mice models suggest DMGV is produced exclusively by AGXT2 catalysed conversion of asymmetric dimethylarginine (ADMA) to DMGV in the kidney and liver [16,44]. AGXT2 mediated conversion of ADMA to DMGV is stimulated by high fat diets, high sucrose diets, low-fibre diets, as well as lack of exercise (Figure 3) [45]. Multivariable models in a landmark study with over 5000 patients demonstrate that the association of DMGV with CAD is independent of ADMA [16]. This suggests that despite upregulation of AGXT2 resulting in increased DMGV production through the degradation of ADMA, a potent inhibitor of endothelial nitric oxide production, the association of DMGV with CAD is not dependent on ADMA [16,46].

The association of the lipid metabolic pathway with CAD that we observed is in keeping with prior studies that have shown that circulating lipid phosphatidylcholine, TMAO, and betaine are associated with increased risk of CVD^41^. The association we identified between the nucleotide metabolic pathway (comprising: adenosine, cAMP, 2′-deoxyadenosine, cytosine, thymidine, and uridine) and CAD is a novel finding. To our knowledge, the metabolites within this pathway have not previously been found to be associated with CAD. Whilst uridine has been associated with acute ischaemic stroke in humans in a metabolomic study, a mechanistic role has not been found. Our novel finding warrants validation and further investigation.

The metabolic associations we report in this study, enriched for metabolites with known biological roles in atherosclerosis, build upon findings from recent studies utilising metabolomics to identify signatures of incident cardiovascular disease that have identified: acylcarnitine, dicarboxylacylcarnitine, TMAO, amino acids such as phenylalanine, glutamate, and several lipid classes, as being associated with cardiovascular disease [15]. One study utilizing an untargeted metabolomic approach in a general population cohort identified four metabolites that predicted cardiovascular events independently of traditional cardiovascular risk factors: lysophosphatidylcholine 18:1, lysophosphatidylcholine 18:2, monoglyceride 18:2, and sphingomyelin 28:1 [47], these metabolites were not assessed in our targeted approach. Another novel study investigated acute markers of myocardial infarction using a targeted metabolomic assessment in patients with hypertrophic cardiomyopathy who were undergoing an alcohol septal ablation procedure that intentionally causes an area of myocardial infarction [47]. They identified a four-metabolite signature of acute myocardial infarction, consisting of: aconitic acid, hypoxanthine, TMAO, and threonine [48].

As we have demonstrated, metabolomics provide a snapshot of metabolic processes and of pathways at a point in time, allowing for metabolite and metabolic pathway associations with disease to be identified. With recent technological, computational and bioinformatic advances, the precision medicine era is upon us and the time is ripe for deep biological phenotyping incorporating standard clinical and health data, together with multi-omic data (including: metabolomics, genomics, proteomics, lipidomics, transcriptomics, and immunophenotyping), to unravel novel disease pathways, improve risk prediction, and provide personalised preventative therapies [49,50]. The BioHEART study is well placed to contribute to this important work with ongoing multi-platform imaging and biological phenotyping underway and additional samples securely stored in a biobank for future analyses [11]. In addition to baseline clinical data, outcome data is also being collected through traditional methods including follow up phone call interviews and medical record reviews, together with data linkage with electronic health data and administrative datasets [11]. This will allow the currently assessed metabolite data to be assessed for associations with outcome data in the future.

This study had a number of strengths, of particular note the detailed CAD phenotyping utilising CTCA segmental scores to identify and quantify plaque composition and burden including sub-clinical CAD. There were, however, also some limitations. Risk factors were treated as binary based on patient reported clinical diagnosis, and we recognise that SMuRFs—particularly hypertension and dyslipidaemia—are continuous and a gradient of risk that may not be adequately captured by diagnostic cut offs [51]. Participants in this study had a clinically indicated CTCA investigating on a wide variety of symptoms and risk factors. The broader applicability of our findings to a primary prevention screening population is not known. We selected four metabolites to be assessed in a candidate manner based on associations in previous unbiased metabolomic analyses. Whilst each of these markers were associated with one or more CAD measure in our study, only DMGV retained statistical significance when assessed in an unbiased manner, controlling for FDR 0.05, highlighting the strength of the association between DMGV and the specific coronary artery disease phenotypes.

## 5. Conclusions

A not-insubstantial burden of CAD is not well explained by traditional risk factors. We report novel associations between specific CAD plaque phenotypes and four metabolites with known roles in atherosclerosis or CAD events, and identified two metabolic pathways strongly associated with CAD, independent of traditional risk factors. These pathways warrant further investigation at both a biomarker and mechanistic level.

## Figures and Tables

**Figure 1 cells-10-00980-f001:**
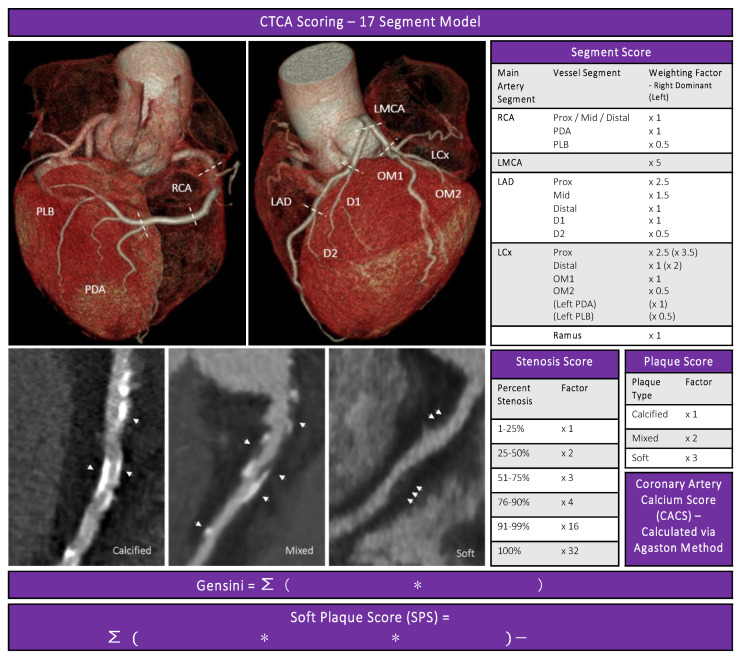
BioHEART-CT semi-quantitative plaque scores. Incorporating the validated Gensini score and the addition of a plaque composition multiplier to obtain the soft plaque score. Right coronary artery (RCA), posterior descending artery (PDA), posterolateral branch (PLB), left main coronary artery (LMCA), left anterior descending artery (LAD), left circumflex artery (LCx), diagonal branch 1,2 (D1, D2), obtuse marginal branch 1,2 (OM1, OM2). Arrows indicate coronary artery plaque.

**Figure 2 cells-10-00980-f002:**
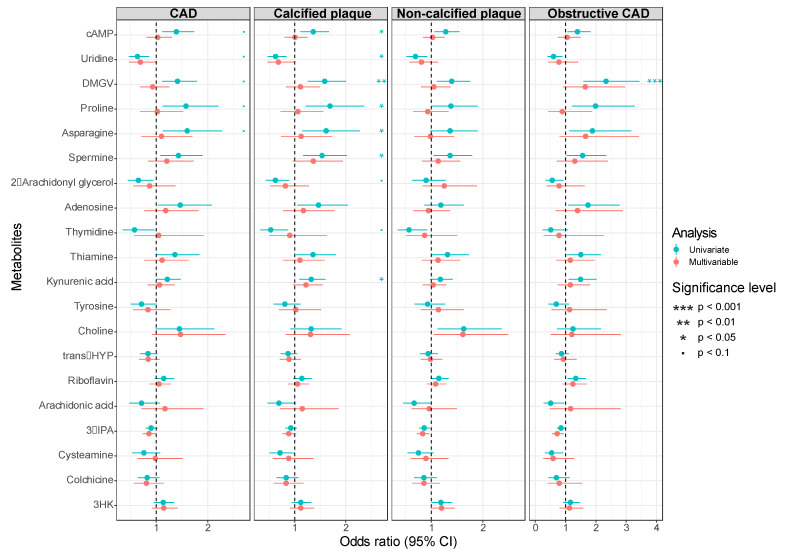
Odds Ratios for association of top 20 ranked metabolites for presence of CAD, calcified plaque, soft plaque, and obstructive CAD. Ranked according to strength of univariate association with presence of CAD. *p* values are adjusted for a 5% false discovery rate using Benjamini and Hochberg approach: *p* < 0.1, * *p* < 0.05, ** *p* < 0.01, *** *p* < 0.001. Turquoise represents univariate associations. Red represents associations adjusted for age, sex, hypertension, hypercholesterolaemia, diabetes, and significant smoking.

**Figure 3 cells-10-00980-f003:**
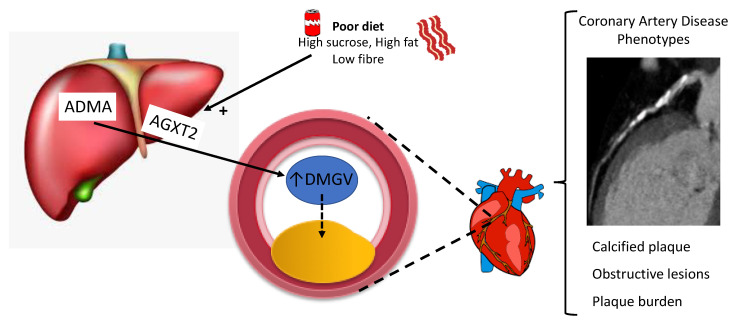
Schematic representation of pathways leading to increased alanine glyoxylate aminotransferase II (AGXT2) stimulated conversion of asymmetric dimethylarginine (ADMA) to dimethylguanidino valerate (DMGV) in the liver and kidneys (not shown). Increased plasma DMGV is associated with specific coronary artery disease phenotypes.

**Table 1 cells-10-00980-t001:** BioHEART-CT Clinical Characteristics and CAD Scores.

	Whole Cohort *n* = 1002	CAD Present *n* = 656)	No CAD *n* = 346	Calcified Plaque Present *n* = 593	No Calcified Plaque *n* = 409	Soft Plaque Present *n* = 540	No Soft Plaque *n* = 462	Obstructive CAD *n* = 181	No CAD *n* = 346
Age-years, median (IQR)	62 (53–70)	65 (58–72)	53 (46–60)	66 (59–72)	54 (46–62)	66 (59–72)	56 (47–64)	60 (63–75)	53 (46–60)
Female, n (%)	447 (44.6)	297 (45.3)	150 (43.4)	266 (44.9)	181 (44.3)	249 (46.1)	198 (42.9)	83 (45.9)	150 (43.4)
BMI-kg/m2, mean (SD)	26.9 (4.8)	27.1 (4.8)	26.5 (4.9)	27.1 (23.8–29.8)	26.6 (4.8)	27.0 (4.7)	26.8 (5.0)	26.9 (4.7)	26.5 (4.9)
Hypertension, n (%)	390 (38.9)	296 (45.1)	94 (27.2)	278 (46.9)	112 (27.4)	244 (45.2)	146 (31.6)	95 (52.5)	94 (27.2)
Diabetes Mellitus, n (%)	87 (8.7)	66 (10.1)	21 (6.1)	63 (10.6)	24 (5.9)	55 (10.2)	32 (6.9)	22 (12.2)	21 (6.1)
Significant Smoking History, n (%)	207 (20.7)	164 (25)	43 (12.4)	155 (26.1)	52 (12.7)	143 (26.5)	64 (13.9)	60 (33.1)	43 (12.4)
Hypercholesterolaemia, n (%)	600 (59.9)	436 (66.5)	164 (47.4)	408 (68.8)	192 (46.9)	360 (66.7)	240 (51.9)	122 (67.4)	164 (47.4)
SMURFs:									
0, n (%)	223 (22.3)	104 (15.9)	119 (34.4)	81 (13.7)	142 (34.7)	82 (15.2)	141 (30.5)	22 (12.2)	119 (34.4)
1, n (%)	419 (42.8)	266 (40.5)	153 (44.2)	240 (40.5)	179 (43.8)	223 (41.3)	196 (42.4)	65 (35.9)	153 (44.2)
2, n (%)	266 (26.5)	205 (31.3)	61 (17.6)	194 (32.7)	72 (17.6)	165 (30.6)	101 (21.9)	62 (34.3)	61 (17.6)
3, n (%)	80 (8.0)	67 (10.2)	13 (3.8)	64 (10.8)	16 (3.9)	57 (10.6)	23 (5.0)	29 (16.0)	13 (3.8)
4, n (%)	14 (1.4)	14 (2.1)	0 (0)	14 (2.4)	0 (0)	13 (2.4)	1 (0.2)	3 (1.7)	0 (0)
Significant Family History CAD, n (%)	205 (20.5)	135 (20.6)	70 (20.2)	118 (19.9)	87 (21.3)	111 (20.6)	94 (20.3)	37 (20.4)	70 (20.2)
Oral anticoagulant, n (%)	88 (8.8)	69 (10.5)	19 (5.5)	62 (10.5)	26 (6.4)	55 (10.2)	33 (7.1)	19 (10.5)	19 (5.5)
Antiplatelet, n (%)	175 (17.5)	128 (19.5)	47 (13.6)	120 (20.2)	55 (13.4)	108 (20.0)	67 (14.5)	44 (24.3)	47 (13.6)
Statin, n (%)	330 (32.9)	268 (40.9)	62 (17.9)	259 (43.7)	71 (17.4)	227 (42.0)	103 (22.3)	85 (47.0)	62 (17.9)
Beta Blocker, n (%)	142 (14.2)	108 (16.5)	34 (9.8)	100 (16.9)	42 (10.3)	86 (15.9)	56 (12.1)	40 (22.1)	34 (9.8)
ACEi or ARB, n (%)	316 (31.5)	247 (37.3)	69 (19.9)	235 (39.6)	81 (19.8)	206 (38.1)	110 (23.8)	80 (44.2)	69 (19.9)
Calcium Channel Blocker, n (%)	102 (10.2)	84 (12.8)	18 (5.2)	79 (13.3)	23 (5.6)	70 (13.0)	32 (6.9)	25 (13.8)	18 (5.2)
Diuretics, n (%)	72 (7.2)	58 (8.8)	14 (4.0)	51 (8.6)	21 (5.1)	50 (9.3)	22 (4.8)	18 (9.9)	14 (4.0)
CACS, median (IQR)	9.9 (0–146.0)	76 (11.1–314.5)	0 (0-0)	99.3 (23.5–371.4)	0 (0–0)	98.6 (18.3–371.9)	0 (0–0)	405.3 (124.7–967.7)	0 (0-0)
Gensini, median (IQR)	3.5 (0–11.5)	8.5 (4–17.4)	0 (0-0)	9.5 (4.5–18.5)	0 (0-0)	10.0 (5.0–19.0)	0 (0–0)	23.5 (17.5–34.8)	0 (0-0)
SPS, median (IQR)	2.0 (0–9.5)	6.5 (2.5–14.0)	0 (0-0)	6.5 (2.0–14.8)	0 (0-0)	8.5 (4.5–16.9)	0 (0–0)	20.5 (12.0–28.0)	0 (0-0)
No CAD (Gensini = 0), n (%)	346 (34.5)	0	346 (100)	0 (0)	346 (84.6)	0 (0)	116 (25.1)	0 (0)	346 (100)

CAD, indicates coronary artery disease; IQR, interquartile range; BMI, body mass index; SMuRFs, standard modifiable cardiovascular risk factors; ACEi, angiotensin converting enzyme inhibitor (ACEi); ARB, angiotensin receptor II blocker; CACS, coronary artery calcium score; SPS, soft plaque score.

**Table 2 cells-10-00980-t002:** Candidate metabolite associations.

	Univariate Binary Regressions			Multivariable Binary Regressions			Univariate Linear Regressions			Multivariable Linear Regressions		
	OR	95% CI	*p* Value	OR	95% CI	*p* Value	ß Coeff	95% CI	*p* Value	ß Coeff	95% CI	*p* Value
**DMGV**												
Gensini	1.41	1.12 1.79	0.004	0.93	0.69 1.26	0.641	0.3	0.14 0.45	<0.001	0.17	0.02 0.32	0.026
CACS	1.59	1.26 2.01	0	1.11	0.83 1.5	0.482	0.49	0.11 0.87	0.011	0.32	−0.03 0.67	0.074
SPS	1.4	1.11 1.75	0.004	1.05	0.8 1.38	0.717	0.15	−0.02 0.32	0.089	0.06	−0.12 0.23	0.523
Obstructive	2.33	1.59 3.43	<0.0001	1.65	0.92 2.96	0.094						
**TMAO**												
Gensini	0.85	0.62 1.16	0.305	0.83	0.57 1.22	0.35	−0.16	−0.39 0.07	0.185	−0.21	−0.42 −0.01	0.044
CACS	0.98	0.72 1.32	0.885	1.03	0.71 1.5	0.88	−0.46	−0.99 0.08	0.093	−0.58	−1.05 −0.1	0.017
SPS	0.72	0.53 0.97	0.029	0.65	0.46 0.92	0.016	−0.05	−0.29 0.19	0.707	−0.09	−0.32 0.15	0.475
Obstructive	0.79	0.51 1.22	0.292	0.84	0.44 1.59	0.60						
**Phenylala-nine**												
Gensini	0.99	0.65 1.48	0.943	0.85	0.51 1.44	0.556	0.33	0.04 0.62	0.025	0.26	0.01 0.52	0.046
CACS	1.12	0.75 1.67	0.57	0.97	0.59 1.62	0.921	0.88	0.23 1.53	0.008	0.77	0.19 1.35	0.01
SPS	0.88	0.6 1.3	0.526	0.73	0.46 1.15	0.176	0.29	−0.01 0.58	0.058	0.24	−0.05 0.53	0.106
Obstructive	1.84	1.01 3.33	0.046	1.22	0.51 2.91	0.66						
**Glutamate**												
Gensini	1.15	0.84 1.57	0.398	0.96	0.65 1.41	0.819	0.14	−0.09 0.36	0.234	0.1	−0.1 0.3	0.333
CACS	1.21	0.89 1.65	0.214	1.07	0.73 1.57	0.736	0.05	−0.48 0.58	0.863	0.05	−0.43 0.53	0.833
SPS	1.48	1.09 2.01	0.011	1.42	1.00 2.03	0.053	0.12	−0.12 0.36	0.326	0.08	−0.16 0.31	0.529
Obstructive	1.25	0.78 2.01	0.361	1.30	0.64 2.66	0.472						

OR, indicates odds ratio; CI, confidence interbal; coeff, coefficient; DMGV, Dimethylguanidino valerate; CACS, coronary artery calcium score; SPS, soft plaque score; TMAO, Trimethylamine N-oxide.

## Data Availability

The data presented in this study are available on request from the corresponding author. The data are not publicly available due to privacy and ethics requirements.

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
