# Peer review of "Metabolic Signatures in Coronary Artery Disease: Results from the BioHEART-CT Study"

_cells, 2021, doi:10.3390/cells10050980_

Round 1

Reviewer 1 Report

The present study by Vernon et ala describes the association of metabolic molecules and CAD. I think the article is interesting and I have only some minor comments.

Abstract:

calcified plaque, and obstructive CAD (P<0.05 for all). Should be for both?

In unbiased analysis assessing 53 metabolites and controlling for 5% false discovery rate, DMGV was strongly associated with the presence of calcified plaque, OR 1.59 (95% CI 1.26-2.01, P=0.006), obstructive CAD, OR 2.33 (95% CI 1.59-3.43, P=0.0009), and amount of CAD ß-coefficient 0.3 (95% CI 0.14-0.45, P=0.014). The lipid and nucleotide metabolic pathways were associated with presence of CAD, adjusted for traditional risk factors in multivariate analyses. Should be revised to be more accurate and clear

Introduction:

…factors or CAD. We lack biomarker… Revision might improve the flow of the text?

Omics studies utilise large amounts of data aiming to capture a comprehensive rep- resentation of a particular biological parameter (e.g. metabolites) within a biological fluid or tissue and examine the data for relationship to phenotype or disease12. Metabolites are influenced by environmental factors, genetic factors and interactions between the two13. Metabolomics, the systematic identification and quantification of small molecules (up to 1.5 kDa) in biological fluids, thus reflects the metabolic state of the biological system at a point in time14. Based on the sample type analysed, the metabolome can be assessed at a cellular, tissue, organ, or whole- organismal level.  I think this is not essential for the introduction.

We first assessed four candi- date metabolites based on a priori knowledge of their association to cardiovascular/met- abolic diseases. We then assessed associations with metabolic pathways, and finally as- sessed 53 metabolites in a targeted metabolomic analyses in 1002 patients (Supplementary Table 13).  This information belongs to materials and methods and results

Results:

Table 2. the layout should be revised.

Reviewer 2 Report

In the study "Metabolic signatures in coronary artery disease: results from the BioHEART-CT" authors aimed to assess for independent associations between plasma metabolites and specific CAD plaque phenotypes.The study is well designed and conducted and the results clearly presented 
Only one minor point,  the authors should be add alimitations of the study 

Reviewer 3 Report

Dear Editor,

Work sent for review entitled Metabolic signatures in coronary artery disease: results from the BioHEART-CT study it is very interesting and it concerns this medical problem.

Introduction written clear.

Purpose precisely defined.

Material and methods please specify the amount included in the survey of respondents.

Please determine whether all tests were carried out on one CT and specify the manufacturer.

Results Please explain all abbreviations under Table 1 Table 2 is difficult to read.

Interestingly written discussion.
